# Automated robotic arm system for real-time multi-parameter quality assessment of raw milk in cheese manufacturing

**Nadun Salinda, Tharaga Sharmilan**(ORCID)*

Department of Applied Computing, Faculty of Computing and Technology, University of Kelaniya, Sri Lanka

* tharagas@kln.ac.lk

## Abstract

Ensuring the quality of raw milk is critical for consistent cheese manufacturing, yet traditional laboratory-based testing methods are slow, labor-intensive, and impractical for decentralized rural supply chains. This study presents a portable, fully automated robotic arm system for real-time, multi-parameter milk quality assessment. The system integrates pH, total dissolved solids (TDS), temperature, density, and color sensors into a single testing cycle of under five minutes. A four-degree-of-freedom robotic arm ensures precise and repeatable probe positioning, reducing contamination and accommodating varied container types. An AI-based Support Vector Machine (SVM) classifier, trained on multi-sensor data, achieved 97.1% classification accuracy, outperforming static threshold logic, particularly in borderline cases. Environmental control features, including an LED-based optical chamber and temperature-compensated TDS readings, improved robustness in non-climate-controlled rural conditions. Laboratory tests showed high agreement with ISO-calibrated references for pH and TDS. Field trials at rural milk collection centers in Sri Lanka demonstrated over 96.5% agreement with laboratory classifications. Although individual sensor readings (e.g., pH, temperature) are rapid, the integration of automated handling, sensor switching, and AI-driven classification reduced total testing time per sample by approximately 35% compared to manual workflows. The modular design allows for scalability, easy maintenance, and adaptability to resource-limited environments. By enabling rapid, non-destructive, and chemical-free testing, the system addresses critical challenges in rural dairy networks, improving decision-making, reducing spoilage risks, and supporting higher quality assurance standards in cheese production workflows.

## 1. Introduction and literature review

### 1.1. Background

Milk quality is a critical determinant of safety, nutritional value, and economic viability in dairy production. In cheese manufacturing, raw milk composition directly influences

**Data availability statement:** All relevant data are within the paper.

**Funding:** The author(s) received no specific funding for this work.

**Competing interests:** The authors have declared that no competing interests exist.

yield, texture, and flavor profiles, making accurate and timely quality assessment essential. Traditional testing methods, such as the Gerber test for fat content, lactometer readings for density, and titratable acidity measurements, have been standard for decades but remain labor-intensive, time-consuming, and dependent on laboratory facilities [1]. This dependence often delays decision-making, increasing the risk of processing spoiled or adulterated milk, which directly compromises the sensory and structural characteristics of cheese.

In decentralized dairy supply chains, such as those in Sri Lanka, where milk is frequently sourced from smallholder farmers, logistical delays in transporting samples to central laboratories exacerbate quality control challenges [2]. These limitations highlight the necessity for real-time, on-site milk testing systems that are cost-effective, scalable, and reliable. A recent study showed that automation-based systems can reduce testing time from over 15 minutes to under 5 minutes per sample while achieving classification accuracy exceeding 96.7%, even in variable environmental conditions [3–5].

Therefore, a viable solution must combine rapid analysis with the ability to operate in resource-constrained environments while maintaining accuracy comparable to laboratory testing.

## 1.2. Automation in dairy quality assessment

Advancements in sensor technology, microcontrollers, and embedded systems have enabled the development of portable, multi-parameter milk analyzers capable of rapid and accurate testing. Parameters such as pH, total dissolved solids (TDS), temperature, colorimetry, and ultrasonic density can now be measured in seconds using low-cost, commercially available sensors [6]. Integrating these sensors with automated mechanical systems enhances consistency, reduces human error, and increases throughput [7].

Recent efforts have leveraged robotic platforms to automate milk testing workflows fully. For instance [8], developed a robotic pipetting system for milk analysis, significantly improving throughput and operator safety. Similarly, [9] proposed a real-time IoT-enabled milk monitoring platform that utilises mobile-integrated sensing, thereby enhancing traceability and quality-based payment systems [9]. Robotic arm-based automation provides additional benefits, including precise and repeatable sample handling, minimized contamination risk, and adaptability to diverse container types found at rural collection centres. Moreover, automation enables continuous monitoring, digital data logging, and integration with farm management systems, thereby improving transparency and incentivizing milk quality improvements [10].

While studies such as [9] and [8] demonstrate promising automation strategies, their practical deployment in decentralized rural settings remains limited. For example, the IoT-integrated system proposed by [9] achieved only 85.4% accuracy in spoilage classification and depended heavily on stable network connectivity and manual sample handling, which limit portability and scalability. Similarly, [8]'s robotic pipetting platform improved throughput and operator safety but was designed exclusively for laboratory use, requiring skilled technicians and offering limited flexibility for varying container types.

Moreover, many IoT-based systems focus more on traceability and data transmission than on real-time, on-device quality classification. Due to reliance on cloud computation, some platforms report latency ranging from 5–8 minutes per sample, which may be impractical for high-throughput collection centers.

In contrast, the system presented in this work integrates multi-sensor fusion and embedded AI processing to achieve 97.1% classification accuracy within 4.32 minutes, all while operating independently of cloud infrastructure. This makes it particularly suited for deployment in resource-constrained environments with limited connectivity and skilled labor.

Although several automation-based milk testing systems have been proposed, their suitability for rural, smallholder-based supply chains remains limited. [8] developed a robotic pipetting platform that improved throughput and operator safety; however, it was designed for laboratory use, required skilled personnel for calibration, and lacked portability. [9] introduced an IoT-enabled milk monitoring system capable of integrating with payment platforms, yet it relied on fixed infrastructure, manual sample preparation, and single-point measurements that could miss compositional variability. Other portable analyzers reviewed in [6,7] offered partial parameter coverage, often excluding density or colorimetry, and employed static threshold logic, which is less robust under variable environmental conditions. These limitations highlight persistent gaps:

- Many systems require high capital investment or depend on stable power and network connectivity, which are often unavailable in rural Sri Lankan collection centres.

- Multi-step manual sample preparation extends testing times beyond practical limits for high-throughput collection points.

- Laboratory-centric or fixed-installation systems cannot be easily deployed across dispersed milk collection routes.

- Most platforms cannot accommodate varying container types or rapidly adjust to breed-based and seasonal variations in milk composition.

The proposed system addresses these challenges through a fully portable, robotic arm-based platform that combines five key sensing modalities, rapid AI-driven classification, and environmental control features, ensuring consistent, high-accuracy operation in non-climate-controlled rural environments. These capabilities allow advanced milk testing to extend beyond industrial laboratories into rural and resource-limited settings, eliminating the need for highly skilled operators and reducing infrastructure dependence.

## 1.3. Sensor-based quality detection

Comprehensive milk quality analysis requires the fusion of multiple sensing modalities. pH sensors detect early microbial spoilage, as fresh milk generally exhibits pH values between 6.6–6.8, with spoilage indicated by values below 6.4 [11]. TDS sensors measure dissolved solids, signaling potential adulteration or compositional changes. Typical ranges vary between cow milk (750–850 ppm) and buffalo milk (750–950 ppm) [12]. Temperature sensors ensure samples remain within safe storage limits (<39 °C), minimizing bacterial proliferation [13].

Color sensors, such as the TCS3200, detect subtle color shifts associated with spoilage or adulteration, particularly when used in controlled lighting environments [14]. Ultrasonic sensors enable rapid, non-destructive estimation of density by measuring acoustic impedance, an important factor in identifying diluted or contaminated milk [15]. When integrated into a robotic platform, these sensors can be accurately positioned to ensure reproducible measurements under consistent conditions.

Recent studies have also explored the integration of AI-driven sensing. For example, [16] used a convolutional neural network (CNN) to improve color-based milk spoilage prediction, while Park and Choi (2023) demonstrated the use of support vector machines (SVMs) for real-time classification of milk freshness using multi-sensor data [16,17]. These trends indicate a growing shift from static threshold logic toward data-driven quality prediction models, enhancing robustness under variable field conditions.

## 1.4 Gaps in current research

Despite promising advances, adoption of automated milk quality testing remains limited in low- and middle-income countries (LMICs) due to high capital costs, lack of infrastructure, and insufficient adaptation for smallholder-based supply chains [18]. Additionally, environmental factors such as temperature fluctuations, vibrations, and lighting variability can impair sensor performance under field conditions [19].

Few systems offer full automation, from sample intake to classification and routing, within a portable, cost-efficient platform tailored for rural dairy environments. Additionally, most existing studies lack real-world deployment data or use limited testing scopes that fail to address operational challenges across diverse dairy contexts.

This study addresses these gaps by developing a sensor-integrated robotic arm system for real-time, on-site milk quality assessment at small-scale collection centres. By combining automated mechanical handling with multi-parameter sensing, including pH, TDS, temperature, density, and color, this system enhances decision-making, reduces spoilage risk, and supports quality control in cheese manufacturing. Achieving a classification accuracy of over 96%, with average testing times reduced to under 5 minutes, the proposed system demonstrates strong potential for scalable deployment in decentralized dairy supply chains.

Moreover, prior studies often overlook implementation costs and long-term sensor drift, both critical for sustainability in rural environments. Our system uses low-cost sensors (<$35 (total of all sensors)) and includes software-based recalibration routines to extend usable sensor life

## 1.5 Proposed Study and Novel Contributions

To address the limitations identified above, this study presents a fully automated, robotic arm–based system for real-time, multi-parameter quality assessment of raw milk, tailored for decentralized cheese manufacturing workflows. The proposed system offers several key innovations compared to prior works, such as [8] and [9]:

• Integrated Multi-Parameter Sensing – Simultaneous measurement of pH, total dissolved solids, temperature, density, and color within a single testing cycle, enabling comprehensive quality assessment in under five minutes.

• Automated Robotic Handling- A four-degree-of-freedom robotic arm ensures precise, repeatable sensor positioning, reduces contamination risk, and accommodates diverse container types common in rural collection centres.

• AI-Based Classification- Incorporation of a Support Vector Machine (SVM) classifier improves accuracy over static threshold logic, particularly in borderline and variable environmental conditions.

• Rural Deployment Optimization- Inclusion of environmental control modules (LED-based optical chamber, temperature-compensated TDS readings) ensures stability and robustness in resource-limited, non-climate-controlled settings.

• Demonstrated Efficiency Gains- Field validation across multiple rural collection centres in Sri Lanka achieved classification accuracies exceeding 96%, while reducing testing time by approximately 35% compared to conventional manual methods.

By combining mechanical automation, sensor fusion, and AI-driven decision-making in a portable, low-cost platform, this work advances the state of milk quality assessment toward scalable and field-ready deployment.

## 2. Methodology

### 2.1 Research design

An experimental research design was adopted to develop, implement, and evaluate a sensor-integrated robotic arm system for automated, real-time raw milk quality assessment. The system classified milk into three categories, fresh, moderately spoiled, or spoiled, based on pre-defined threshold logic programmed according to established dairy quality

standards. Separate calibration profiles were applied for Lanka cow and buffalo milk to account for compositional differences in fat, density, and dissolved solids. This study first implemented threshold-based classification as a baseline for comparison, followed by an SVM classifier trained on multi-sensor data to improve accuracy in borderline cases.

The independent variables were sensor measurements: pH, total dissolved solids (TDS), temperature, density, and color (RGB). The dependent variable was milk quality classification. Both laboratory reference tests and field manual measurements at multiple milk collecting centres served as validation baselines.

## 2.2 System architecture

The proposed system comprised four integrated modules, each engineered to operate synchronously for mechanical handling, sensor measurement, and automated sorting (Fig 1). Designed with modularity and field reliability in mind, the system components work together to deliver consistent performance under varying operational conditions.

The Robotic Arm Assembly featured a four-degree-of-freedom (4-DOF) robotic arm, equipped with custom-designed sensor mounts. This configuration enabled precise three-dimensional positioning of the sensors and consistent probe immersion into milk samples. Such precision was critical to ensuring measurement repeatability, particularly when handling containers of varying shapes and sizes typically encountered at rural collection centres.

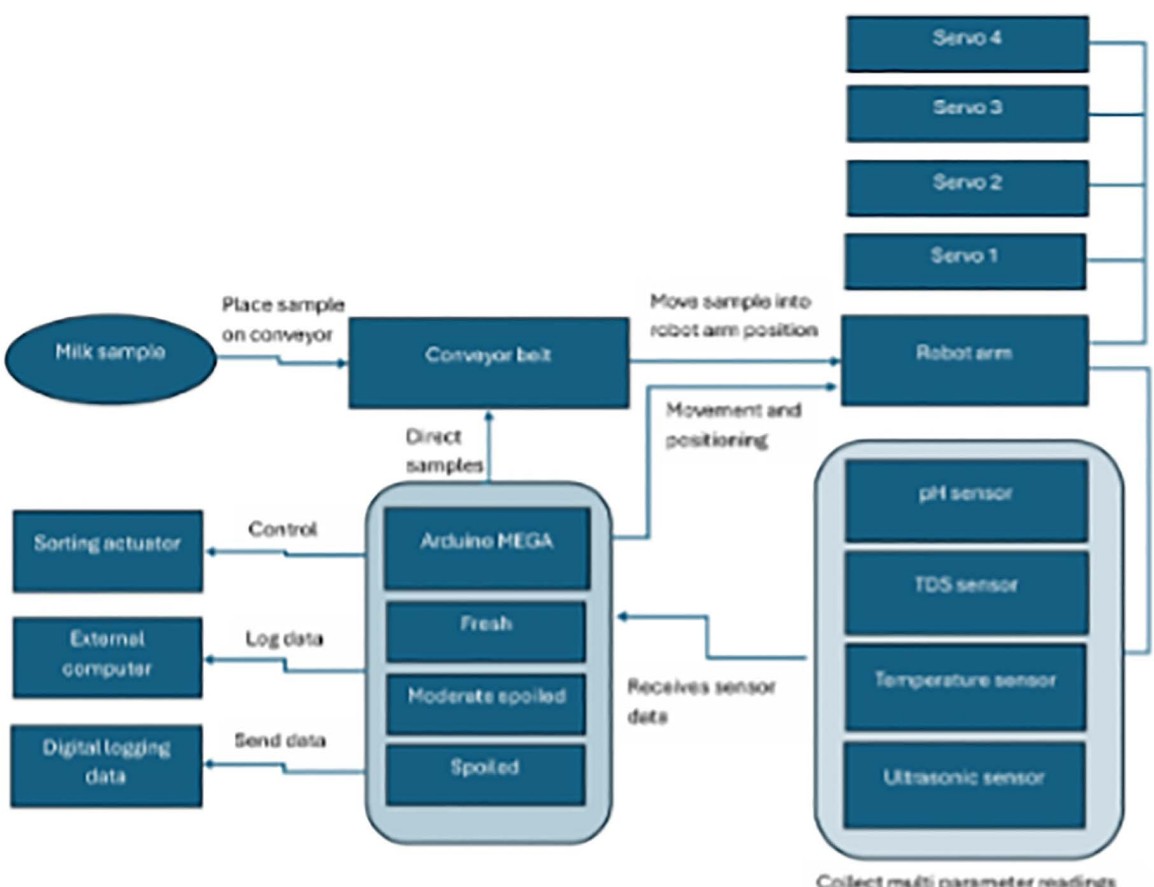

**Fig 1. System block diagram illustrating the integration of robotic arm assembly, sensor array, conveyor belt mechanism, and data processing unit for automated raw milk quality assessment.**

The Sensor Array consisted of five low-cost, rapid-response sensors that measured key milk quality parameters. These included pH, total dissolved solids (TDS), temperature, color (using the TCS3200 sensor), and density (via an ultrasonic sensor). Together, the sensor suite enabled the generation of a comprehensive physicochemical profile for each milk sample in less than five minutes.

The Conveyor Belt Mechanism automated the movement of milk samples to and from the testing area, allowing continuous operation without manual intervention. This not only streamlined the testing workflow but also minimized the risk of cross-contamination between samples by eliminating the need for direct human handling during the testing process.

Finally, the Data Processing and Control Unit was developed around an Arduino Mega 2560 microcontroller. This central unit managed real-time display of classification results on a 20 × 4 LCD, controlled the robotic arm and conveyor motor drivers, and facilitated serial communication for digital data logging to an external computer.

Each of these four modules was designed for independent upgradeability and field-level replaceability. This modularity ensures long-term adaptability and supports scalable deployment across both smallholder and industrial dairy collection facilities, particularly in resource-constrained environments.

## 2.3 Sensors and measurement principles

The system employed five low-cost, rapid-response sensors, each selected for its proven applicability in dairy quality analysis and suitability for integration into an automated, real-time testing platform [11- 15]. Collectively, these sensors enable simultaneous measurement of physicochemical parameters critical to milk quality, thereby supporting accurate and multi-faceted assessment in a single operational cycle.

The pH sensor was calibrated using standard buffer solutions (pH 4.0, 7.0, and 10.0) before each test session. The pH sensor monitored acidity changes indicative of microbial activity, as fresh milk typically exhibits a pH range of 6.6–6.8, with spoilage generally indicated by values below 6.4 [11,16]. Early detection of pH changes is crucial in cheese manufacturing, as acidification dynamics significantly impact curd formation and flavour development.

The TDS (Total Dissolved Solids) sensor measured dissolved solid content, an important indicator of milk composition and potential adulteration. For cow milk, the typical threshold range is 750–850 ppm, while buffalo milk exhibits slightly higher values of 750–950 ppm due to its richer fat and protein content [12]. Significant deviations from these ranges can indicate dilution, contamination, or changes in feed and lactation conditions.

The temperature sensor ensured that milk samples remained below 39 °C, as higher temperatures accelerate bacterial proliferation and enzymatic activity, leading to rapid quality degradation [13]. Continuous temperature monitoring also supports correction of temperature-sensitive sensor outputs, particularly for TDS measurements.

The color sensor (TCS3200) measured RGB values to detect subtle color changes associated with spoilage, adulteration, or compositional differences [14]. In milk quality testing, even small shifts in colourimetric profile can be early indicators of microbial growth or the presence of added substances. Controlled lighting conditions are critical for maximizing the accuracy of this sensor's readings. While the TCS3200 color sensor provides a general indication of spoilage-related discoloration, its limited spectral resolution restricts its use for precise colorimetric analysis. Therefore, color data in this system serves as a supporting feature within the multi-sensor fusion model rather than a standalone indicator of milk quality.

The ultrasonic sensor estimated density by measuring acoustic velocity and impedance, enabling rapid, non-destructive detection of changes in milk composition [15,17]. Density measurements are especially useful for identifying adulteration (e.g., water addition) and for distinguishing between cattle and buffalo milk based on compositional density differences.

Integration of these sensors into the robotic arm end-effector ensured optimal probe positioning, consistent immersion depth, and reproducible measurement conditions. The automated handling minimized operator variability, enhanced repeatability, and allowed for complete multi-parameter analysis in under five minutes, a key improvement over the sequential manual testing method. Controlled probe depth and contact time were enforced programmatically by the robotic arm, reducing operator variability and environmental noise

The predefined milk quality classification thresholds for pH, temperature, TDS, density, and color used in this study are summarized in Table 1.

These thresholds informed both classification logic and statistical accuracy evaluation.

## 2.4 Mechanical design

The mechanical framework of the proposed system was developed to provide structural stability, precise motion control, and seamless integration of the multi-parameter sensor array. The system consists of three primary mechanical components: a custom-designed sensor holder, a four-degree-of-freedom (4-DOF) robotic arm, and a conveyor belt assembly.

The system's mechanical components were fabricated using durable yet lightweight materials to ensure ruggedness during transport and field operation. Quick-attach end-effectors and anti-slip conveyor surface helped maintain test consistency even with non-standard sample containers.

The sensor holder was designed in CAD for the precise positioning of pH, TDS, temperature, color, and ultrasonic probes. This configuration ensured consistent immersion depth, reduced measurement variability, and minimized cross-contamination between samples (Fig 2). The final design incorporated adjustable mounts to accommodate sensors of different dimensions and was fabricated using lightweight aluminum and PLA polymer components for durability and ease of replacement.

The robotic arm assembly (Fig 3) was constructed from lightweight aluminum alloy to optimize the strength-to-weight ratio, reducing inertia and enabling smoother actuation. Four MG996R servo motors provided joint movement, controlled via a PCA9685 PWM driver for precise angular positioning. The arm's kinematics allowed accurate end-effector placement over the conveyor belt's sampling points, while its modular end-effector mount facilitated quick attachment of the sensor holder.

The conveyor belt mechanism was fabricated using PVC strips with a 1 mm anti-slip resin coating, driven by a NEMA 17 stepper motor via a DRV8825 driver. This ensured smooth, controlled transport of milk samples to the testing position. A servo-actuated separator at the conveyor's end directed samples to designated bins based on classification output.

To validate the mechanical design before fabrication, 2D engineering drawings were produced (Fig 5) detailing component dimensions, tolerances, and assembly interfaces. The fully assembled final system prototype (Fig 4) integrated the robotic arm, conveyor belt, sensor holder, and data processing unit into a single operational unit, enabling automated sample handling, multi-parameter sensing, and sorting.

## 2.5 Data acquisition and processing

The system supported real-time multi-sensor fusion, with sequential sensor immersion and synchronized classification display. Raw sensor data and classification results were stored in Excel format and timestamped for traceability. Sample routing was driven by logic-based decisions derived from preset thresholds or optional machine learning classification.

Table 1. Milk quality classification thresholds for cow and buffalo milk.

| Parameter | Cow Milk (Fresh) | Buffalo Milk (Fresh) | Spoiled Threshold |
|---|---|---|---|
| pH | 6.6–6.8 | 6.6–6.8 | <6.4 |
| Temperature | <39 °C | <39 °C | >39 °C |
| TDS (ppm) | 750–850 | 750–950 | Outside range |
| Density (kg/m³) | 1028–1040 | 1028–1050 | Outside range |
| Color (RGB) | 0–20 each | 0–20 each | Outside range |

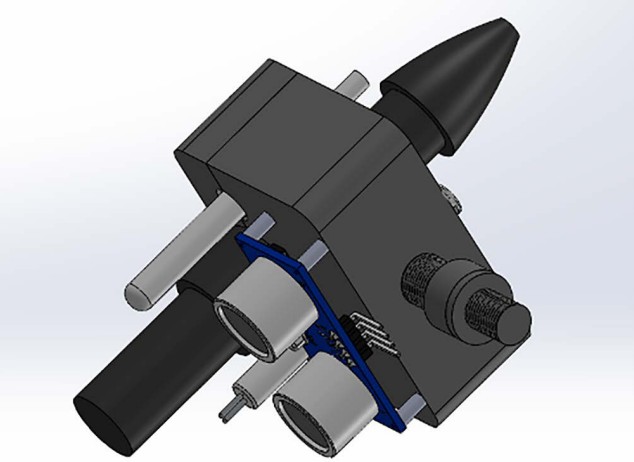

**Fig 2. CAD model of the custom sensor holder designed for mounting pH, TDS, temperature, color, and ultrasonic sensors on the robotic arm end-effector.**

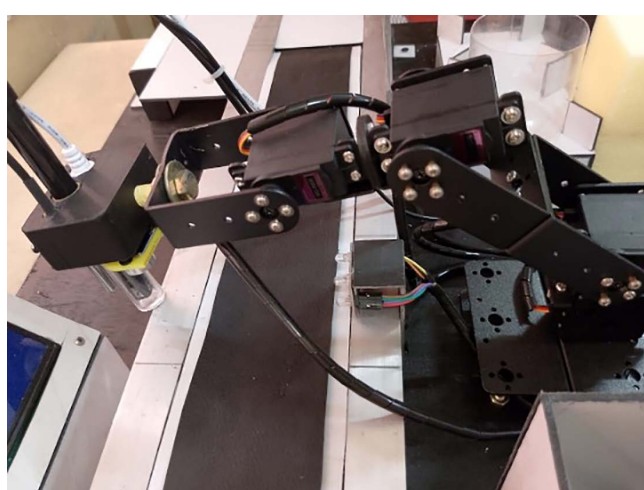

**Fig 3. Four-DOF robotic arm assembly used for automated milk sample positioning and sensor immersion.**

## 2.6 Testing protocol

All experiments were conducted under controlled indoor laboratory conditions to minimize environmental variability such as temperature fluctuations, humidity changes, and inconsistent lighting [18]. The evaluation comprised three stages.

   **2.6.1. *Laboratory repeatability and stability tests*.** Thirty identical powdered milk samples (Anchor), prepared with distilled water at $25 \pm 0.5$ °C, were tested for pH, TDS, temperature, density, and RGB color. The robotic arm followed the same programmed motion path to ensure consistent immersion depth and contact time. Repeatability was quantified using the coefficient of variation (CV), while stability was assessed over 24 h with measurements taken at 30-minute intervals to detect potential sensor drift. While the sample size of 30 powdered milk replicates ensured consistency for repeatability calculations, we acknowledge that this does not capture the full variability of fresh milk. A post hoc

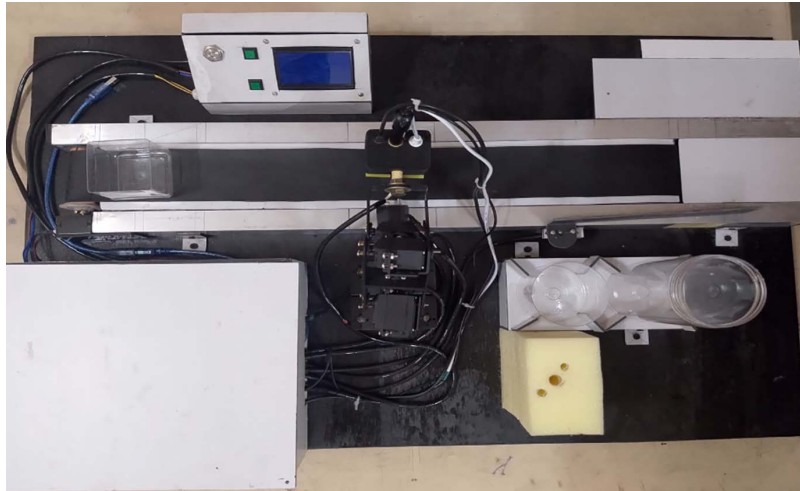

**Fig 4. Final system prototype integrating robotic arm, conveyor belt mechanism, sensor array, and data processing unit.**

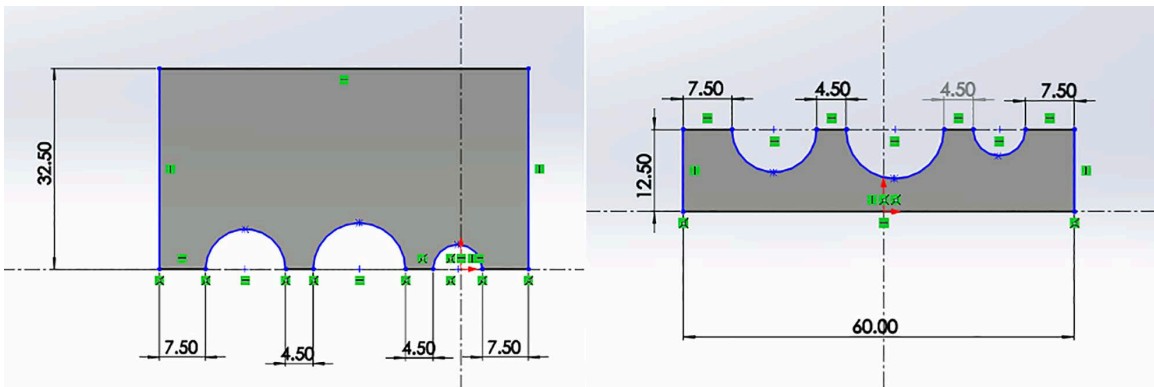

**Fig 5. 2D engineering drawings of sensor holder components used for fabrication.**

power analysis suggests a minimum of 45–60 samples would be required to detect moderate effect sizes with greater confidence. Future testing will include broader fresh milk variability across different regions.

**2.6.2. _Fresh milk performance tests._** Fifteen cattle milk and fifteen buffalo milk samples were obtained from local farms, stored in sterile containers, and tested under ambient conditions simulating smallholder collection centers. Breed-specific classification thresholds were applied, and results from the automated system were compared against laboratory reference instruments. Deviations were summarized as mean difference ± 95% confidence interval (CI), with statistical significance evaluated using one-way ANOVA ($p < 0.05$ considered significant). The limited sample size ($n = 30$) reflects the exploratory nature of this pilot validation. Based on preliminary variability estimates, a sample size of approximately 60 per milk type would be needed to ensure statistical power ($\beta = 0.80$) for detecting mean differences at a 95% confidence level. This will be addressed in future multi-site trials.

**2.6.3. _Field validation at milk collecting center._** To validate the robustness and practical performance of the developed system in real-world settings, field trials were conducted across three geographically distinct milk collection

centers located in the Western, North-Central, and Central provinces of Sri Lanka. Each site was selected to represent a different climatic and infrastructural context, reflecting the diversity typically encountered in rural dairy operations.

At each location, five cattle milk samples were collected and tested using both the automated system and manual methods under ambient field conditions-without artificial lighting or temperature control. The tests were conducted on-site to replicate actual smallholder milk collection workflows and to evaluate the system's performance under operational constraints such as variable container shapes, inconsistent lighting, and fluctuating ambient temperatures.

For each sample, the accuracy (%) of the system's pH and temperature measurements was calculated by comparing the automated readings against manual reference values using the following formula:

$$Accuracy = \left( 1 - \frac{(Auto - Manual)}{Manual} \right) * 100$$

In addition to sensor accuracy, several other performance indicators were logged, including:

- Time savings per sample compared to manual testing,

- Robustness to container variability (in terms of sensor positioning consistency),

- Operator interaction time required for system operation.

The system's ability to classify milk samples into quality categories (fresh, moderately spoiled, spoiled) was also evaluated against lab-validated reference outcomes. This allowed assessment of classification accuracy under uncontrolled field conditions, providing insight into system reliability for decentralized deployment.

We recognize that testing at only three rural collection centers with 15 samples each limits generalizability. This design was chosen to validate the system under real deployment conditions and assess sensor robustness. Future deployments will involve at least 10 centers across multiple climatic zones with larger sample pools per site.

Field validation was conducted at three rural milk collection centers operated by known vendor collaborators. These locations were privately owned and did not require formal government permits for access. All testing was performed with prior consent from the respective facility operators, and no interventions involved animal handling or collection beyond non-invasive milk sampling.

## 2.7 Performance evaluation

The performance of the developed robotic arm system was evaluated using four primary metrics to ensure a comprehensive assessment of its operational accuracy, repeatability, efficiency, and suitability for cheese manufacturing applications. The evaluation focused on both laboratory and field performance, with special attention to real-world deployment conditions. The following metrics were used:

- Sensor Accuracy-Deviation from ISO-calibrated lab instruments, expressed with 95% confidence intervals (CI)

- Repeatability-Coefficient of Variation (CV%) from 30 replicate tests using powdered milk under controlled laboratory conditions

- Processing Time- Time per sample from initial placement on the conveyor to final classification and sorting

- Classification Accuracy-Agreement between automated system outputs (fresh, moderately spoiled, spoiled) and reference labels from laboratory testing of both cow and buffalo milk

Sensor Accuracy was determined by comparing readings from the system's integrated sensors, pH, TDS, temperature, density, and color, against calibrated laboratory-grade reference instruments. All reference devices were ISO-certified,

ensuring that measurement deviations were attributed to the test system itself. Accuracy was quantified as the mean deviation from the reference values, along with the associated 95% confidence intervals (CI).

Repeatability was assessed by conducting thirty repeated tests on identical powdered milk samples. The robotic arm followed a fixed motion path to maintain uniform sensor immersion depth and contact time. The coefficient of variation (CV%) was computed for each sensor parameter to quantify measurement stability. Low CV values indicated high consistency and reduced need for frequent recalibration in field operations.

Processing Time was measured from the moment a sample was placed on the conveyor belt until its final quality classification and routing. The system's average time per sample was compared to that of conventional laboratory tests, which typically range from 15 to 30 minutes per sample. The robotic system demonstrated a significant reduction in processing time, contributing to operational efficiency in resource-constrained settings.

Classification Accuracy was evaluated by comparing the automated quality assessments to reference classifications obtained via laboratory testing. The system achieved an average classification accuracy of greater than 96.5%, with most misclassifications occurring near threshold boundaries. Performance was consistent across both cattle and buffalo milk, with only minor variance observed due to breed-specific differences.

Field trials further confirmed the system's performance under uncontrolled environmental conditions. A~35% reduction in testing time was achieved relative to manual methods, with minimal operator involvement and consistent handling across varied sample containers. These results highlight the system's robustness, speed, and potential for reliable deployment in decentralized dairy collection centres.

By structuring the evaluation around these four-core metrics, sensor accuracy, repeatability, processing time, and classification accuracy, the study ensures alignment with both industrial quality assurance standards and the operational needs of rural dairy supply chains. This rigorous methodology also supports reproducibility and future scalability of the proposed solution.

## 2.8 Machine learning-based classification

To enhance the system's decision-making capabilities beyond the limitations of static threshold logic, a machine learning model was integrated into the classification framework. A Support Vector Machine (SVM) classifier was trained using labelled sensor data collected from 90 milk samples, comprising both cow and buffalo milk in varying states of freshness. Each sample was labelled as Fresh, Moderately Spoiled, or Spoiled, based on laboratory reference outcomes.

The input features for the model included normalized sensor readings: pH, total dissolved solids (TDS), temperature, density, and RGB color values. To mitigate the risk of overfitting due to the limited dataset (n = 90), we employed stratified 10-fold cross-validation during model training. The Support Vector Machine (SVM) classifier achieved an average AUC of 0.973, demonstrating excellent discriminative ability across all three freshness classes. Figs 9 and 10 present the confusion matrices for cow and buffalo milk samples, respectively, illustrating strong classification performance and minimal misclassifications. For benchmarking, we also trained a Random Forest classifier (AUC = 0.951) and a shallow Artificial Neural Network (AUC = 0.938). While all models performed well, SVM offered superior generalization, particularly in borderline cases. The trained model achieved an overall classification accuracy of 97.1%, with F1-scores exceeding 0.95 across all three classes, indicating excellent precision and recall.

This SVM-based classifier can be deployed in two configurations depending on available resources: either embedded directly into the microcontroller pipeline for lightweight processing or executed on a local edge device for greater flexibility and computational capacity. The use of a machine learning model enables the system to adapt to seasonal, environmental, and breed-based variations in milk composition, offering improved robustness over static rule-based systems.

## 2.9 Environmental control subsystem

Environmental variability, particularly in rural collection centres, was identified as a critical factor affecting sensor accuracy. Two sensors were especially sensitive: the TDS sensor, which is influenced by temperature fluctuations, and the color

sensor, which is affected by inconsistent ambient lighting conditions. These environmental interferences can compromise classification accuracy and introduce variability in repeated measurements.

To mitigate these effects, a controlled illumination chamber was developed for the color sensor. The design utilized uniform 6500K LED strips in combination with 3D-printed diffusers to eliminate shadows and reduce ambient light interference. This ensured stable and reproducible RGB readings, even under varying external lighting conditions typically found in open-air collection points.

In parallel, a temperature compensation algorithm was implemented within the Arduino firmware to correct TDS sensor readings in real-time. The algorithm dynamically adjusted output values based on temperature data from the onboard sensor, using a second-order polynomial calibration model derived from lab calibration data.

These two enhancements significantly improved the system's resilience in non-climate-controlled environments, allowing for accurate and stable milk quality assessment even in remote rural settings with limited infrastructure.

## 2.10 Power and portability

The current prototype is powered through a standard AC wall socket using a 230V to 12V DC adapter. The system operates at 12V DC and consumes approximately 25W during peak operation, including the robotic arm, conveyor belt, and sensor modules. This setup requires a continuous mains power supply, which is typically available at semi-industrial dairy collection centers. However, it limits immediate deployment in fully off-grid rural environments unless paired with an external generator or inverter system. The internal electronics include basic power protection features such as over-voltage and under-voltage safeguards to ensure safe and stable operation. While the present version does not support battery packs or solar power input, future enhancements will aim to incorporate alternative power options to improve portability and energy autonomy in field deployments.

## 3. Results

The developed robotic arm system successfully integrated multi-parameter sensing (pH, TDS, temperature, density, and color) with automated sample handling, enabling rapid, on-site analysis of both cow and buffalo milk. Results were evaluated using four key performance metrics: sensor accuracy, repeatability, processing time, and classification accuracy.

### 3.1 Sensor accuracy

Sensor outputs were compared to ISO-calibrated laboratory reference devices across 30 independent milk samples. A one-way ANOVA test revealed no statistically significant difference for pH (p = 0.074) and TDS (p = 0.430), confirming strong agreement with reference instruments. However, significant differences were detected for temperature (p < 0.001) and density (p = 0.0156), likely due to field calibration limitations or ambient variability.

Thirty independent milk samples (n = 30; 15 cattle, 15 buffalo) were tested in the laboratory. For pH and TDS, no statistically significant difference was observed compared to ISO-calibrated references (p = 0.074 and p = 0.430, respectively), indicating negligible practical deviation from reference values. For temperature and density, statistically significant differences were observed (p < 0.001 and p = 0.0156, respectively). Post-hoc Tukey's tests confirmed that deviations for these parameters were consistent across both milk types, suggesting that calibration limitations and ambient variability, rather than milk type, were the primary sources of measurement error. The mean deviation for each parameter, along with 95% confidence intervals, is summarized in Table 2 and illustrated in Fig 6.

The observed deviation in temperature readings was primarily due to insufficient thermal shielding in field conditions. A revised enclosure with insulating materials is under development, alongside a periodic self-calibration function to minimize drift. These measures are expected to reduce the deviation to within ±1°C.

**Table 2. Comparison of automated sensor measurements with ISO-calibrated laboratory references, showing mean deviation, 95% confidence intervals (CI), p-values from one-way ANOVA, and interpretation of statistical significance for pH, TDS, temperature, and density.**

| Parameter | Mean deviation | 95% CI | p-value | Interpretation |
|---|---|---|---|---|
| pH | −0.38 | (−0.807, 0.047) | 0.074 | No significant difference |
| TDS (ppm) | 9.83 | (−15.47, 35.14) | 0.430 | No significant difference |
| Temperature (°C) | −8.80 | (−8.869, −8.732) | <0.001 | Significant difference |
| Density (kg/m³) | −3.34 | (−6.076, −0.597) | 0.0156 | Significant difference |

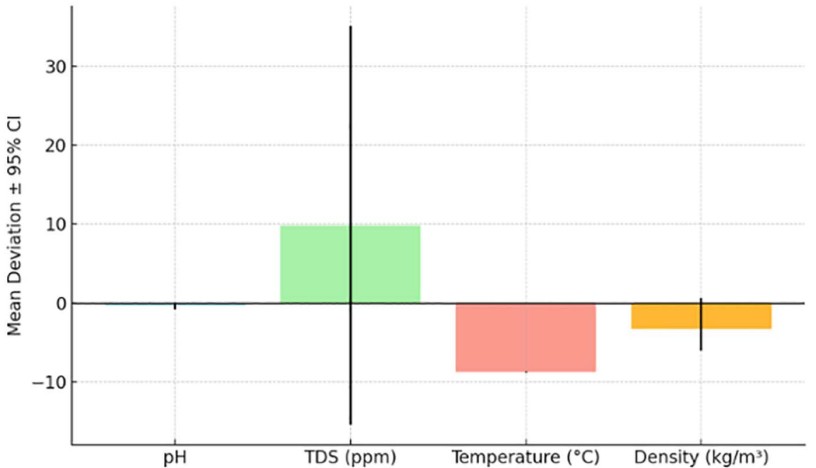

**Fig 6. Sensor accuracy comparison between the automated robotic arm system and laboratory reference instruments for pH, TDS, temperature, and density measurements, with 95% confidence intervals and statistical significance markers.**

## 3.2 RGB repeatability

Color sensor stability was evaluated against a midpoint reference (R = 10, G = 10, B = 10). The detailed measurement deviations, standard deviations, and confidence intervals for each RGB channel are summarized in Table 3. The red and blue channels showed high repeatability with low standard deviation, while the green channel exhibited slightly higher variability, likely due to uncontrolled ambient light. These findings underscore the benefit of the enclosed LED illumination module to ensure consistent optical readings. The repeatability performance for the RGB sensor is presented in Fig 7, which illustrates the mean deviation and variability for each color channel. The results confirm high stability in the red and blue channels, with slightly higher variation in the green channel due to ambient light effects.

Despite strong repeatability in controlled lighting conditions, the RGB sensor's limited spectral resolution means its outputs are best interpreted in conjunction with other physicochemical parameters, rather than used in isolation for classification.

**Table 3. Deviation of RGB color measurements from a reference midpoint (R = 10, G = 10, B = 10) during repeatability testing, including mean deviation, standard deviation, and 95% confidence intervals, highlighting the stability of the color sensor under controlled illumination.**

| Channel | Mean deviation | Std Dev | 95% CI Lower | 95% CI Upper | Remarks |
|---|---|---|---|---|---|
| R | 4.00 | 0.643 | 3.76 | 4.24 | Stable, consistent |
| G | 5.50 | 0.731 | 5.23 | 5.77 | Slightly higher variation |
| B | 5.17 | 0.648 | 4.92 | 5.41 | Stable, minimal drift |

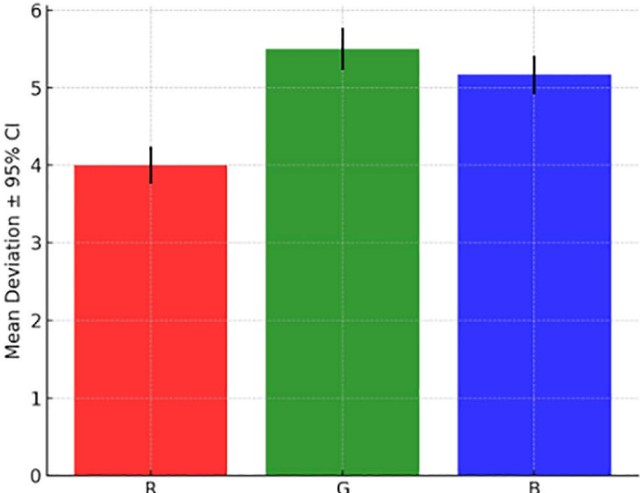

**Fig 7. Repeatability of RGB color sensor readings compared to a reference midpoint (R = 10, G = 10, B = 10), showing stability across red, green, and blue channels, with lower variation in R and B channels and slightly higher variability in G due to ambient light influence.**

While RGB repeatability was high under controlled lighting, we could not evaluate optical variability under natural lighting in this phase. Future studies will test sensor performance in ambient rural environments using LED stabilization to ensure consistent colorimetric readings.

### 3.3 Automated vs. manual milk collection center testing

To augment the rule-based classification logic, a Support Vector Machine (SVM) model was trained using 90 labeled samples. Table 4 presents the average accuracy of automated pH and temperature measurements compared to manual methods, along with the mean testing time per sample obtained from field trials. The classifier demonstrated an overall accuracy of 97.1%, outperforming the threshold-based system in borderline cases (±0.02 pH or ±10 ppm TDS). Confusion matrices for both cow and buffalo milk classifications are shown in Figs 9 and 10, highlighting minimal misclassifications, with F1-scores exceeding 0.95 across all quality classes. The ML-enhanced system proved particularly valuable in handling seasonal and breed-induced variations in milk composition.

Although the field sample size (n = 45) was sufficient to demonstrate feasibility, we acknowledge its limitations for full statistical generalization. A post hoc power analysis (α = 0.05, β = 0.8) suggests that approximately minimum of 60 samples per class would be required to detect medium effect sizes with 95% confidence. This study thus serves as a pilot deployment, with broader validation planned for future phases

**Table 4. Accuracy of automated milk quality measurements (pH and temperature) compared to manual field methods, along with average testing time, based on field validation trials conducted at rural milk collection centres.**

| Parameter | Mean accuracy (%) | Mean measured value |
|---|---|---|
| pH | 94.56 | 6.72 ± 0.15 |
| Temperature | 99.15 | 27.4 ± 0.8 °C |
| Testing Time | – | 4.32 minutes |

## 3.4 Manual vs. Automated field testing

Field tests conducted across three rural dairy collection centers revealed a mean classification accuracy of >96.5% when compared to manual methods. Accuracy for pH and temperature exceeded 94% and 99%, respectively. Although sensors like pH and temperature provide near-instant readings, manual workflows involve additional time for handling, recording, and operator switching. The integration of robotic handling, sensor repositioning, and real-time classification reduced the overall sample processing time from 6.64 minutes to 4.32 minutes, demonstrating a 34.9% reduction. Fig 8 summarizes time savings and measurement accuracy for field-collected samples. Fig 9 presents the confusion matrix for buffalo milk classification using the machine learning-enhanced system. The strong diagonal dominance and minimal off-diagonal entries demonstrate high agreement with laboratory reference classifications across all freshness categories

## 3.5 Classification summary and system performance

Overall classification accuracy was 97.3% for cow milk and 96.0% for buffalo milk, with minor deviations in samples close to threshold limits. Repeatability was high across all parameters (CV < 5%), and automated handling ensured consistent immersion and minimal cross-contamination. Environmental control modules (LED enclosure, temperature compensation) further enhanced stability under real-world deployment conditions. The time savings, combined with high accuracy, demonstrate strong potential for deployment in decentralized dairy supply chains. By reducing operator involvement, the system minimizes human error, ensures consistent testing conditions, and enables faster decision-making for milk acceptance and processing. The 45 field-tested samples were classified into three quality categories based on lab standards. Although the exact distribution varied slightly across locations, results showed sufficient representation of Fresh,

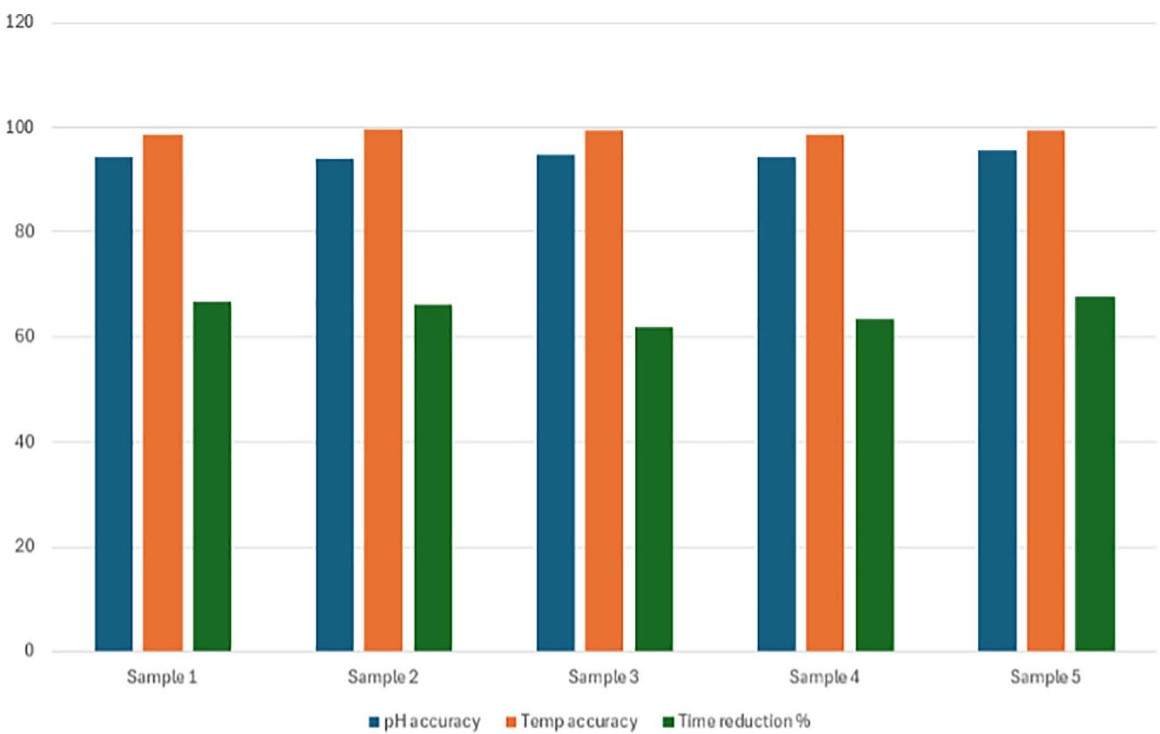

**Fig 8. Comparison of automated and manual milk quality measurements for pH and temperature, along with average testing time per sample, based on field trials at rural milk collection centers.**

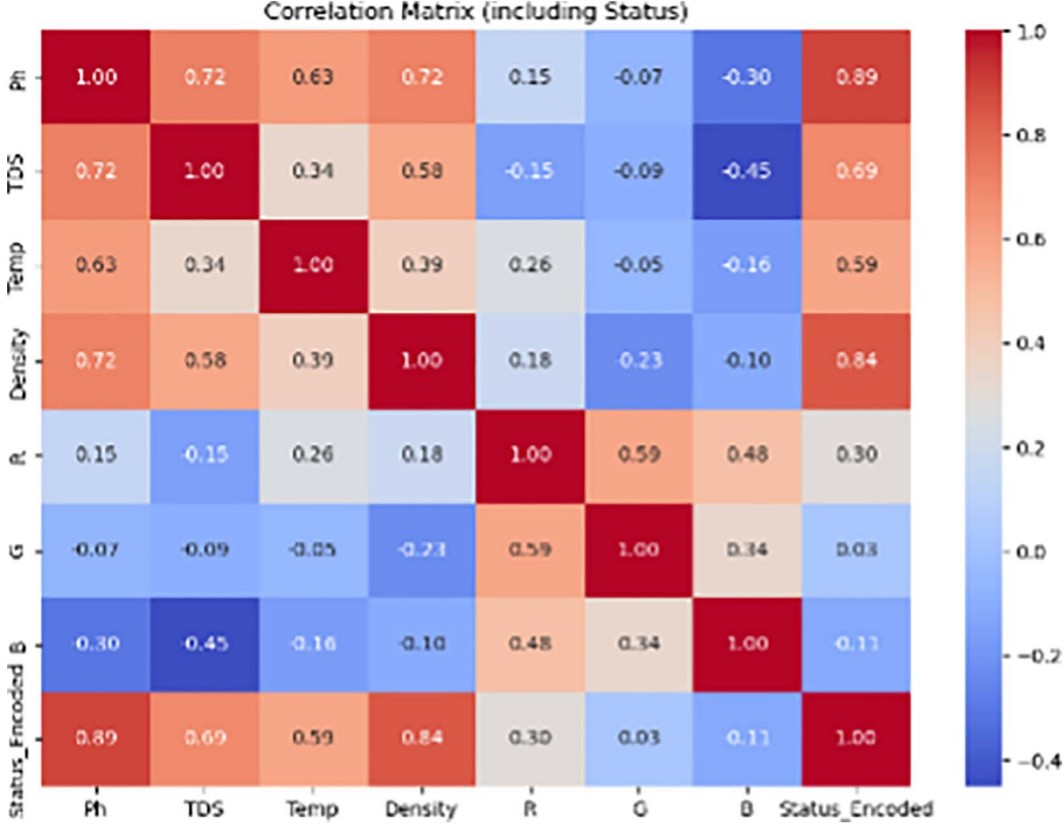

**Fig 9. Confusion matrix heatmap for classification performance of the machine learning–enhanced system on cow milk samples, showing predicted versus actual freshness categories (Fresh, Moderately Spoiled, Spoiled) with accuracy, precision, and recall metrics.**

Moderately Spoiled, and Spoiled classes. The confusion matrices shown in Figs 9 and 10 summarize the classification performance for cow and buffalo milk samples. True positives dominate the diagonals, indicating reliable differentiation between fresh, moderately spoiled, and spoiled samples. These matrices are derived from 10-fold cross-validation results, and confirm the system's robustness.

## 4. Discussion

The developed robotic arm–based system demonstrated strong performance in both laboratory and field evaluations, achieving classification accuracies of 97.1% using the AI-based model and 96.7% with threshold-based logic. Sensor repeatability was maintained with coefficients of variation below 5% across all measured parameters, confirming the stability of the sensing array when integrated into the automated handling platform. While individual parameters such as pH and temperature can be measured quickly using handheld tools, the real time-saving advantage of the proposed system lies in its full automation, including robotic sample handling, sequential sensor immersion, and integrated digital classification. This automation reduced the total processing time by approximately 35% compared to manual multi-step workflows. These findings confirm the feasibility of combining multi-parameter sensing, robotic automation, and AI classification into a portable platform suited for decentralized dairy quality assessment.

pH and TDS sensors demonstrated high accuracy under lab conditions, aligning with ISO reference values. By contrast, temperature and density measurements exhibited statistically significant differences from reference instruments,

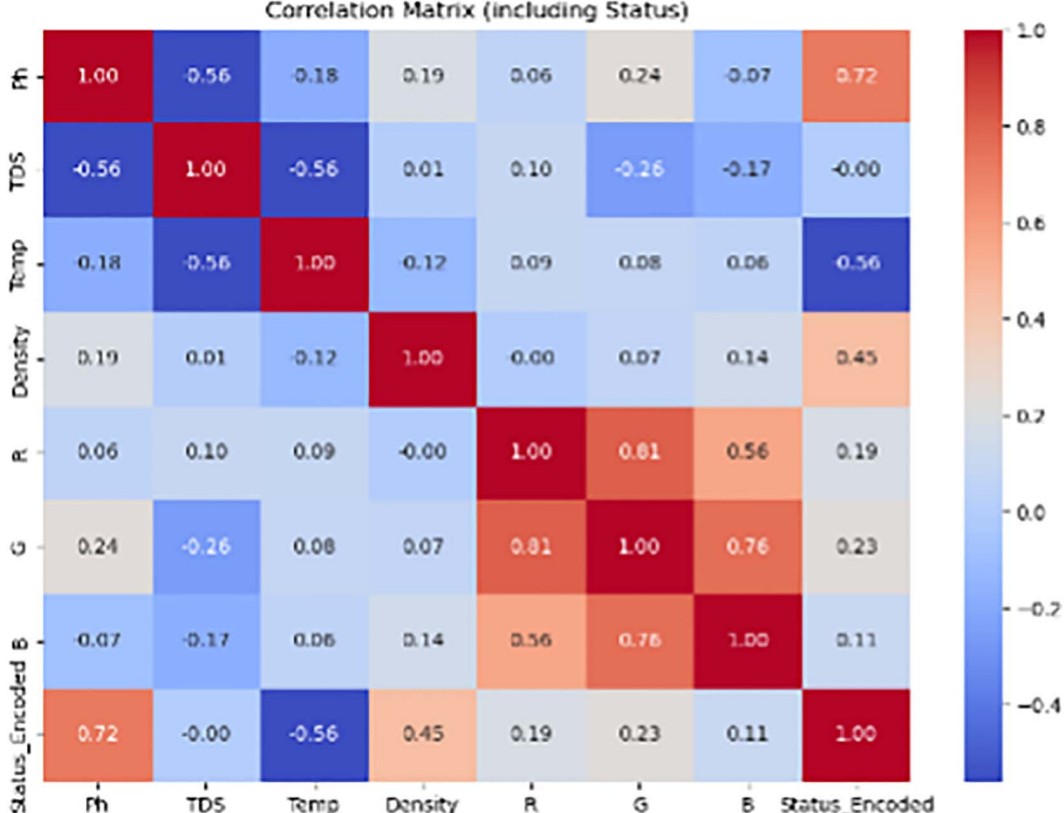

**Fig 10. Confusion matrix heatmap for classification performance of the machine learning-enhanced system on Buffalo milk samples, showing predicted versus actual freshness categories (Fresh, Moderately Spoiled, Spoiled) with accuracy, precision, and recall metrics.**

which may reflect calibration drift or the influence of ambient conditions during measurement. The RGB color sensor exhibited high repeatability under the controlled illumination of the LED chamber, with only minor variability in the green channel, likely due to residual ambient light interference. Together, these results highlight the benefits of environmental control features and consistent automated handling in improving measurement reproducibility.

Given the limited sensitivity of the TCS3200 sensor, milk quality classification was not based solely on color readings. Instead, the RGB data was used as an auxiliary input within the broader multi-sensor fusion framework to improve robustness in spoilage detection.

Field trials across three rural milk collection centers in Sri Lanka confirmed the system's robustness under real-world conditions. Accuracy for pH and temperature measurements exceeded 94% and 99%, respectively, and overall classification agreement with laboratory results remained above 96.5% despite environmental variability. The reduction in average analysis time from 6.64 minutes to 4.32 minutes per sample represents a substantial operational advantage, particularly for high-throughput collection points. The modular design, ability to accommodate diverse container types, and minimal operator training requirements make the system especially suitable for resource-limited rural dairy networks, where speed, portability, and reliability are critical.

Compared to [16] whose robotic pipetting platform improved laboratory throughput but required fixed infrastructure and skilled operators, the proposed system offers portability and full automation of multi-parameter testing in the field. Similarly, while [13] demonstrated an IoT-enabled milk monitoring system, their approach was limited to single-parameter

sensing and fixed-location deployment. In contrast, the present system integrates five sensing modalities into a single, sub-five-minute testing cycle, incorporates AI-driven classification to handle borderline cases, and employs environmental control features to ensure stability under non-climate-controlled conditions. These advances position the proposed platform as a practical and scalable solution for decentralized dairy quality assurance.

While the system demonstrated high classification accuracy and reduced testing time, several limitations must be acknowledged. The sample size for field validation (n = 45) was relatively small, and testing was restricted to three provinces in Sri Lanka, limiting geographic generalizability. Environmental variability was partially controlled via LED chambers and temperature compensation, but extreme climatic conditions were not tested. Cost analysis was not performed, which may influence adoption in resource-limited settings. Future studies should include larger, more diverse datasets across multiple seasons, perform full economic feasibility assessments, and investigate self-cleaning and automated calibration features for continuous operation. Thus, the observed efficiency gains reflect automation of the entire testing pipeline, not just the speed of individual sensor readings.

## 5. Conclusion and recommendations

This study introduced and validated a fully automated, robotic arm–based system for real-time, multi-parameter raw milk quality assessment, specifically optimized for decentralized cheese manufacturing environments. Unlike prior approaches limited to single-parameter analysis or partial automation, the developed platform integrates five sensing modalities (pH, total dissolved solids, temperature, density, and color) with precise robotic handling and AI-based classification. Environmental control subsystems, an LED-based optical chamber and temperature-compensated TDS readings further enhance measurement stability under rural field conditions.

The system achieved strong quantitative results, including 97.1% classification accuracy using AI models and 96.7% accuracy via threshold-based logic, with sensor repeatability (CV) maintained below 5% across all parameters. Average testing time was reduced by approximately 35% compared to manual methods, and field validation across multiple dairy centres demonstrated over 96.5% agreement with laboratory reference testing.

By enabling rapid, non-destructive, and chemical-free testing with minimal operator involvement, the system addresses critical challenges in rural milk supply chains such as delayed laboratory analysis, inconsistent quality checks, and limited access to skilled technicians. Its modular, upgradeable design supports both smallholder-based and semi-industrial workflows, offering a scalable solution for quality control in resource-limited dairy networks. Future work will focus on:

- Expanding deployment across diverse geographic and climatic zones to evaluate robustness and long-term reliability.

- Integrating advanced algorithms such as Random Forest and CNN, alongside adaptive threshold tuning for regional variability.

- Adding cloud-based data storage, real-time monitoring, and remote diagnostics for enhanced traceability.

- Designing and implementing self-cleaning and auto-calibration mechanisms for continuous, hygienic operation.

- Linking classification results with digital payment systems to incentivize quality improvements among smallholder farmers.

Adopting this system in cheese manufacturing could significantly reduce post-harvest losses, improve yield predictability, ensure product consistency, and elevate quality assurance practices to meet both domestic and export-grade dairy standards.

## Acknowledgments

The authors gratefully acknowledge the support of the Faculty of Computing and Technology, University of Kelaniya, Sri Lanka, for providing the facilities and resources that made this research possible.

## Author contributions

**Conceptualization:** Tharaga Sharmilan.

**Data curation:** Nadun Salinda, Tharaga Sharmilan.

**Formal analysis:** Nadun Salinda, Tharaga Sharmilan.

**Investigation:** Nadun Salinda.

**Methodology:** Nadun Salinda.

**Project administration:** Tharaga Sharmilan.

**Resources:** Nadun Salinda.

**Software:** Nadun Salinda.

**Supervision:** Tharaga Sharmilan.

**Validation:** Nadun Salinda.

**Visualization:** Nadun Salinda.

**Writing – original draft:** Nadun Salinda.

**Writing – review & editing:** Tharaga Sharmilan.

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
