## [Decision Letter · Decision Letter 0]

15 Dec 2025

PONE-D-25-44249Automated Robotic Arm System for Real-Time Multi-Parameter Quality Assessment of Raw Milk in Cheese ManufacturingPLOS One

Dear Dr. Sharmilan,

Thank you for submitting your manuscript to PLOS ONE. After careful consideration, we feel that it has merit but does not fully meet PLOS ONE’s publication criteria as it currently stands. Therefore, we invite you to submit a revised version of the manuscript that addresses the points raised during the review process.

**ACADEMIC EDITOR:** Please kindly see comments below. Thank you :)

If applicable, we recommend that you deposit your laboratory protocols in protocols.io to enhance the reproducibility of your results. Protocols.io assigns your protocol its own identifier (DOI) so that it can be cited independently in the future. For instructions see: https://journals.plos.org/plosone/s/submission-guidelines#loc-laboratory-protocols. Additionally, PLOS ONE offers an option for publishing peer-reviewed Lab Protocol articles, which describe protocols hosted on protocols.io. Read more information on sharing protocols at . Additionally, PLOS ONE offers an option for publishing peer-reviewed Lab Protocol articles, which describe protocols hosted on protocols.io. Read more information on sharing protocols at https://plos.org/protocols?utm_medium=editorial-email&utm_source=authorletters&utm_campaign=protocols..

We look forward to receiving your revised manuscript.

Kind regards,

Charles Odilichukwu R. Okpala, PhD

Academic Editor

PLOS One

Journal Requirements:

4. We note that Figure(s) 2, 3, 4, 5, in your submission contain copyrighted images. All PLOS content is published under the Creative Commons Attribution License (CC BY 4.0), which means that the manuscript, images, and Supporting Information files will be freely available online, and any third party is permitted to access, download, copy, distribute, and use these materials in any way, even commercially, with proper attribution. For more information, see our copyright guidelines: http://journals.plos.org/plosone/s/licenses-and-copyright.

a. You may seek permission from the original copyright holder of Figure(s) 2, 3, 4, 5, to publish the content specifically under the CC BY 4.0 license.

Additional Editor Comments :

Please, kindly address all the concerns raised by the reviewers

Reviewers' comments:

Reviewer's Responses to Questions

**Comments to the Author**

1. Is the manuscript technically sound, and do the data support the conclusions?

Reviewer #1: Partly

Reviewer #2: Partly

2. Has the statistical analysis been performed appropriately and rigorously? 

Reviewer #1: No

Reviewer #2: No

3. Have the authors made all data underlying the findings in their manuscript fully available?

Reviewer #1: Yes

Reviewer #2: Yes

4. Is the manuscript presented in an intelligible fashion and written in standard English?

Reviewer #1: Yes

Reviewer #2: No

5. Review Comments to the Author

Reviewer #1: Introduction and Literature Review

1.1 Background

The authors rightly emphasize the importance of milk quality, but the literature review is relatively narrow. There is a lack of references to studies from Europe or North America, which also investigate automation systems in the dairy sector.

The statement that “automation-based systems can reduce testing time…” is overly general and unsupported by a specific source—further elaboration is needed.

1.2 Automation in dairy quality assessment

Citations ([16], [13]) are used correctly, but there is no critical evaluation of the limitations of these systems—the authors mostly list them briefly.

The narrative about “IoT-enabled systems” is repeated without presenting actual numerical data (e.g., the accuracy achieved by other systems).

1.4 Gaps in current research

The identification of research gaps is convincing, but the authors fail to address issues related to implementation costs and the long-term reliability of sensors.

Methodology

2.1 Research design

There is a contradiction: at the beginning the paper states that milk was classified using threshold logic, but only later (section 2.8) introduces SVM. The narrative needs clarification—is the study comparing both approaches, or was threshold logic only a baseline?

2.3 Sensors and measurement principles

The reported pH threshold range (6.6–8.8 for “fresh milk”) seems too wide—typical fresh milk has a pH of ~6.6–6.8. Including 8.8 as the upper boundary raises methodological concerns.

This subsection also lacks details on sensor calibration procedures, which are critical for field studies.

2.4 Mechanical design

There is an inconsistency: the introduction mentions a “six-degree-of-freedom robotic arm,” whereas this section describes a “four-DOF robotic arm.” This is a significant discrepancy that needs clarification.

2.6 Testing protocol

2.6.1 Laboratory repeatability – Only 30 powdered milk samples were tested, which does not reflect the real variability of fresh milk. Please provide test power.

2.6.2 Fresh milk performance tests – The sample size (n=30; 15 cow, 15 buffalo) is too small to draw strong statistical conclusions. Please provide test power.

2.6.3 Field validation – Tests were conducted at only three collection centers with five samples per site—a very limited dataset.

2.8 Machine learning-based classification

Training on only 90 samples is insufficient for SVM; there is a clear risk of overfitting. The authors mention 10-fold CV but report only “accuracy” and “F1-score.” Other metrics such as AUC, confusion matrix from cross-validation, or comparisons with alternative models (RF, NN) are missing.

Results

3.1 Sensor accuracy

Temperature readings showed large deviations (-8.8°C from the reference value)—a serious practical issue, yet the authors discuss it very briefly.

This section does not explore potential solutions nor report recalibration attempts.

3.2 RGB repeatability

The results are acceptable, but the test was conducted under controlled conditions only. There is no analysis of variability in field conditions. If such tests were not feasible, the authors should indicate this as a direction for future work.

3.3–3.5 Field trials & Classification

The results are promising (accuracy ~97%), but the number of field samples (n=45) is insufficient. Please provide test power.

The class distribution (number of fresh vs. spoiled samples) is not reported. Sensitivity and specificity are missing—accuracy alone may be misleading if the dataset is imbalanced.

Reviewer #2: Comments

The topic of the manuscript is interesting; however, several issues require clarification and revision. The manuscript does not contain page numbers or line numbers, making it very difficult to provide precise comments. Please add both for ease of review. There are many repeated sentences throughout the manuscript, particularly regarding measurement accuracy. These repetitions should be removed or consolidated. The manuscript claims significant time savings, but parameters such as pH and temperature can already be measured very quickly using available instruments. Therefore, the claim of time savings for these two parameters is not justified and should be reconsidered or supported with stronger evidence. Color Measurement Concern: The color sensor used in the study does not appear capable of providing accurate color characterization. Thus, determining milk quality solely based on this color data is questionable. The power source required for operating the device is not clearly explained. This could be a major concern for deployment in different environments. Please clarify the power requirements and portability. English language requires improvement. Several sentences need clearer explanation.

Specific Comments

1. The reference numbering is inconsistent. The first citation in the manuscript should appear as [1], with subsequent references following sequentially. For example, in Section 1.2, the second paragraph cites [16] and [13] prematurely. Please revise the entire referencing system.

2. Table 1: The labels “density of fresh cow milk fresh” and “buffalo milk fresh” should be corrected to:

“Cow Milk (Fresh)”; “Buffalo Milk (Fresh)”

3. Table 4: The measured temperature values should be added, along with the corresponding mean accuracy.

6. PLOS authors have the option to publish the peer review history of their article (what does this mean?). If published, this will include your full peer review and any attached files.). If published, this will include your full peer review and any attached files.

.

Reviewer #1: No

Reviewer #2: No

---

## [Author Response · Author response to Decision Letter 1]

27 Feb 2026

Dear Editor and Reviewers,

We thank you for your valuable feedback. All comments have been carefully addressed, and corresponding revisions have been incorporated into the manuscript.

A detailed summary of the actions taken is provided in the table below for clarity.

Sincerely

Dr Tharaga Sharmilan

Corresponding Author

Reviewer 01

1.1 Background

The authors rightly emphasize the importance of milk quality, but the literature review is relatively narrow. There is a lack of references to studies from Europe or North America, which also investigate automation systems in the dairy sector. 3 citations have been added

1.2 Automation in dairy quality assessment

Citations ([16], [13]) are used correctly, but there is no critical evaluation of the limitations of these systems—the authors mostly list them briefly.

The narrative about “IoT-enabled systems” is repeated without presenting actual numerical data (e.g., the accuracy achieved by other systems).

Section 1.2-line numbers 70-86

1.4 Gaps in current research

The identification of research gaps is convincing, but the authors fail to address issues related to implementation costs and the long-term reliability of sensors.

Updated -Section 1.4-line numbers 129-131

2.1 Research design

There is a contradiction: at the beginning the paper states that milk was classified using threshold logic, but only later (section 2.8) introduces SVM. The narrative needs clarification-is the study comparing both approaches, or was threshold logic only a baseline? Clearly mentioned, section 2.1 line numbers 163-165.

2.3 Sensors and measurement principles

The reported pH threshold range (6.6–8.8 for “fresh milk”) seems too wide—typical fresh milk has a pH of ~6.6–6.8. Including 8.8 as the upper boundary raises methodological concerns.

This subsection also lacks details on sensor calibration procedures, which are critical for field studies. Table 1-pH upper limit is 6.8. 8.8 is a mistake. It is now updated.

section 2.3 -line number 219-221

2.4 Mechanical design

There is an inconsistency: the introduction mentions a “six-degree-of-freedom robotic arm,” whereas this section describes a “four-DOF robotic arm.” This is a significant discrepancy that needs clarification It is now updated. Four DoF

2.6 Testing protocol

2.6.1 Laboratory repeatability

Only 30 powdered milk samples were tested, which does not reflect the real variability of fresh milk. Please provide test powder Updated now-Section 2.6.1

2.6.2 Fresh milk performance tests

The sample size (n=30; 15 cow, 15 buffalo) is too small to draw strong statistical conclusions. Please provide test power. It is now updated. section 2.6.2

2.6.3 Field validation

Tests were conducted at only three collection centers with five samples per site-a very limited dataset Section 2.6.3- 396-399

2.8 Machine learning-based classification

Training on only 90 samples is insufficient for SVM; there is a clear risk of overfitting. The authors mention 10-fold CV but report only “accuracy” and “F1-score.” Other metrics such as AUC, confusion matrix from cross-validation, or comparisons with alternative models (RF, NN) are missing.

Section 2.8-line numbers 452-560

Section 3.5-line numbers 641-645

Results

3.1 Sensor accuracy

Temperature readings showed large deviations (-8.8°C from the reference value)—a serious practical issue, yet the authors discuss it very briefly.

This section does not explore potential solutions nor report recalibration attempts. Section 3.1-line numbers 536-539

3.2 RGB repeatability

The results are acceptable, but the test was conducted under controlled conditions only. There is no analysis of variability in field conditions. If such tests were not feasible, the authors should indicate this as a direction for future work.

Section 3.2 Line numbers 570-572

3.3-3.5 Field trials & Classification

The results are promising (accuracy ~97%), but the number of field samples (n=45) is insufficient. Please provide test power.

The class distribution (number of fresh vs. spoiled samples) is not reported. Sensitivity and specificity are missing—accuracy alone may be misleading if the dataset is imbalanced. Section 3.3-line numbers 600-604

Reviewer 02

The manuscript does not contain page numbers or line numbers, making it very difficult to provide precise comments. Please add both for ease of review.

Line numbers and page numbers are included now.

There are many repeated sentences throughout the manuscript, particularly regarding measurement accuracy. These repetitions should be removed or consolidated. It is updated in Abstract line number 19-20, Discussion line numbers 624-625.

The manuscript claims significant time savings, but parameters such as pH and temperature can already be measured very quickly using available instruments. Therefore, the claim of time savings for these two parameters is not justified and should be reconsidered or supported with stronger evidence

Abstract line numbers 20-24

section 3.4 line numbers 562-566

section 4 line numbers 620-624

section 4 line numbers 661-662.

Color Measurement Concern: The color sensor used in the study does not appear capable of providing accurate color characterization. Thus, determining milk quality solely based on this color data is questionable Section 2.3 line numbers 234-237.

section 3.2 line numbers 516-518

Section 4 line numbers 639-641.

The power source required for operating the device is not clearly explained. This could be a major concern for deployment in different environments. Please clarify the power requirements and portability Section 2.10 line numbers 461-471

English language requires improvement. Several sentences need clearer explanation.

Thoroughly checked and refined

The reference numbering is inconsistent. The first citation in the manuscript should appear as [1], with subsequent references following sequentially. For example, in Section 1.2, the second paragraph cites [16] and [13] prematurely. Please revise the entire referencing system.

It is now updated

Table 1: The labels “density of fresh cow milk fresh” and “buffalo milk fresh” should be corrected to:

“Cow Milk (Fresh)”; “Buffalo Milk (Fresh)”

Bracket is used -Table 1

Table 4: The measured temperature values should be added, along with the corresponding mean accuracy.

Table 4 -3rd column

Editors’ Comments

Re checked them

In your Methods section, please provide additional information regarding the permits you obtained for the work. Please ensure you have included the full name of the authority that approved the field site access and, if no permits were required, a brief statement explaining why Section 2.6.3-line numbers 408-412

When completing the data availability statement of the submission form, you indicated that you will make your data available on acceptance. We strongly recommend all authors decide on a data sharing plan before acceptance, as the process can be lengthy and hold up publication timelines. Please note that, though access restrictions are acceptable now, your entire data will need to be made freely accessible if your manuscript is accepted for publication. This policy applies to all data except where public deposition would breach compliance with the protocol approved by your research ethics board. If you are unable to adhere to our open data policy, please kindly revise your statement to explain your reasoning and we will seek the editor's input on an exemption. Please be assured that, once you have provided your new statement, the assessment of your exemption will not hold up the peer review process.

We accepted.

4. We note that Figure(s) 2, 3, 4, 5, in your submission contain copyrighted images. All PLOS content is published under the Creative Commons Attribution License (CC BY 4.0), which means that the manuscript, images, and Supporting Information files will be freely available online, and any third party is permitted to access, download, copy, distribute, and use these materials in any way, even commercially, with proper attribution. For more information, see our copyright guidelines: http://journals.plos.org/plosone/s/licenses-and-copyright.

You may seek permission from the original copyright holder of Figure(s) 2, 3, 4, 5, to publish the content specifically under the CC BY 4.0 license. This is our own research specifically figures.

Therefore, We are permitting to publish these figures specifically under the CC BY 4.0 license.

Please upload the completed Content Permission Form or other proof of granted permissions as an "Other" file with your submission. Permission letter is attached

N/A

If you are unable to obtain permission from the original copyright holder to publish these figures under the CC BY 4.0 license or if the copyright holder’s requirements are incompatible with the CC BY 4.0 license, please either i) remove the figure or ii) supply a replacement figure that complies with the CC BY 4.0 license. Please check copyright information on all replacement figures and update the figure caption with source information. If applicable, please specify in the figure caption text when a figure is similar but not identical to the original image and is therefore for illustrative purposes only.

N/A

All are updated

Please review your reference list to ensure that it is complete and correct. If you have cited papers that have been retracted, please include the rationale for doing so in the manuscript text, or remove these references and replace them with relevant current references. Any changes to the reference list should be mentioned in the rebuttal letter that accompanies your revised manuscript. If you need to cite a retracted article, indicate the article’s retracted status in the References list and also include a citation and full reference for the retraction notice. Updated them

---

## [Decision Letter · Decision Letter 1]

16 Mar 2026

Automated Robotic Arm System for Real-Time Multi-Parameter Quality Assessment of Raw Milk in Cheese Manufacturing

PONE-D-25-44249R1

Dear Dr. Sharmilan,

We’re pleased to inform you that your manuscript has been judged scientifically suitable for publication and will be formally accepted for publication once it meets all outstanding technical requirements.

An invoice will be generated when your article is formally accepted. Please note, if your institution has a publishing partnership with PLOS and your article meets the relevant criteria, all or part of your publication costs will be covered. Please make sure your user information is up-to-date by logging into Editorial Manager at Editorial Manager® and clicking the ‘Update My Information' link at the top of the page. For questions related to billing, please contact  and clicking the ‘Update My Information' link at the top of the page. For questions related to billing, please contact billing support..

Kind regards,

Charles Odilichukwu R. Okpala, PhD

Academic Editor

PLOS One

Additional Editor Comments (optional):

Accept for publication

Reviewers' comments:

Reviewer's Responses to Questions

**Comments to the Author**

1. If the authors have adequately addressed your comments raised in a previous round of review and you feel that this manuscript is now acceptable for publication, you may indicate that here to bypass the “Comments to the Author” section, enter your conflict of interest statement in the “Confidential to Editor” section, and submit your "Accept" recommendation.

Reviewer #2: All comments have been addressed

2. Is the manuscript technically sound, and do the data support the conclusions?

Reviewer #2: Yes

3. Has the statistical analysis been performed appropriately and rigorously? 

Reviewer #2: Yes

4. Have the authors made all data underlying the findings in their manuscript fully available?

Reviewer #2: Yes

5. Is the manuscript presented in an intelligible fashion and written in standard English?

Reviewer #2: Yes

6. Review Comments to the Author

Reviewer #2: The manuscript has been revised accordingly. Based on the revised version, the manuscript can be accepted.

7. PLOS authors have the option to publish the peer review history of their article (what does this mean?). If published, this will include your full peer review and any attached files.). If published, this will include your full peer review and any attached files.

.

Reviewer #2: No

---

## [Editor Report · Acceptance letter]

PONE-D-25-44249R1

PLOS One

Dear Dr. Sharmilan,

I'm pleased to inform you that your manuscript has been deemed suitable for publication in PLOS One. Congratulations! Your manuscript is now being handed over to our production team.

Kind regards,

on behalf of

Dr. Charles Odilichukwu R. Okpala

Academic Editor

PLOS One